# On the Correctness and Sample Complexity of Inverse Reinforcement Learning

**Abi Komanduru**
Purdue University
West Lafayette IN 47906
akomandu@purdue.edu

**Jean Honorio**
Purdue University
West Lafayette IN 47906
jhonorio@purdue.edu

## Abstract

Inverse reinforcement learning (IRL) is the problem of finding a reward function that generates a given optimal policy for a given Markov Decision Process. This paper looks at an algorithmic-independent geometric analysis of the IRL problem with finite states and actions. A L1-regularized Support Vector Machine formulation of the IRL problem motivated by the geometric analysis is then proposed with the basic objective of the inverse reinforcement problem in mind: to find a reward function that generates a specified optimal policy. The paper further analyzes the proposed formulation of inverse reinforcement learning with $n$ states and $k$ actions, and shows a sample complexity of $O(d^2 \log(nk))$ for transition probability matrices with at most $d$ nonzeros per row, for recovering a reward function that generates a policy that satisfies Bellman's optimality condition with respect to the true transition probabilities.

## 1 Introduction

Reinforcement Learning is the process of generating an optimal policy for a given Markov Decision Process (MDP) along with a reward function. Often, in situations including apprenticeship learning, the reward function is unknown but optimal policy can be observed through the actions of an *expert*. In cases such as these, it is desirable to learn a reward function generating the observed optimal policy. This problem is referred to as Inverse Reinforcement Learning (IRL) Ng & Russel (2000). It is well known that such a reward function is not necessarily unique. The IRL problem can be formulated in two different ways. The first is the form considered in Ng & Russel (2000) as well as this paper, which considers a standard MDP. Several other approaches instead consider the linearly-solvable MDP (LMDP) formulation presented in Dvijotham & Todorov (2010). Various algorithms to solve the IRL problem have been proposed including linear programming Ng & Russel (2000), Hybrid IRL Neu & Szepesvári (2007), Maximum Margin Planning Ratliff et al. (2006), Multiplicative Weights for Apprenticeship Learning Syed et al. (2008) and Bayesian estimation Ramachandran & Amir (2007). Approaches to the LMDP formulation of IRL include Maximum Entropy IRL Ziebart et al. and Gaussian Process IRL Levine et al. (2011). Methods such as Abbeel & Ng (2004) looked at using IRL to solve the apprenticeship learning problem by trying to find a reward function that maximizes the margin of the expert's policy. However, none of the prior works provide a formal guarantee that the reward function obtained from the empirical data is optimal for the true transition probabilities in inverse reinforcement learning. We note that the objective for the LMDP formulation of IRL is different than that of the standard MDP formulation. Indeed, given the true transition probabilities (although in practice we only access estimated probabilities), methods using the standard MDP formulation recover a policy that is Bellman optimal while this may not be the case for methods using the LMDP formulation.

This paper looks at formulating the IRL problem by using the basic objective of inverse reinforcement: to find a reward function that generates a specified optimal policy. The paper also looks at establishing a sample complexity to meet this basic goal when the transition probabilities are estimated from observed trajectories. To achieve this, an algorithmic-independent geometric analysis of the IRL problem with finite states and action is provided. A L1-regularized Support Vector Machine (SVM) formulation of the IRL problem motivated by the geometric analysis is then proposed. The formulation provided is a nonparametric approach as compared to approaches that use features derived from states, such as Neu & Szepesvári (2007) and Ziebart et al.. Theoretical analysis of the sample complexity of the L1 SVM formulation is then performed. Finally, experimental results comparing the L1 SVM formulation to the linear-programming method presented in Ng & Russel (2000) as well as several other standard and LMDP methods are presented showing the improved performance of the L1 SVM formulation with respect to the basic objective, i.e Bellman optimality with respect to the true transition probabilities. To the best of our knowledge, we are the first to provide an algorithm with formal guarantees for *inverse* reinforcement learning.

## 2 Preliminaries

The formulation of the IRL problem is based on a standard Markov Decision Process (MDP) $(S, A, \{P_{sa}\}, \gamma, R)$, where

- $S$ is a finite set of $n$ states.
- $A = \{a_1, \ldots, a_k\}$ is a set of $k$ actions.
- $P_a \in [0,1]^{n \times n}$ are the state transition probabilities for action $a$. We use $P_a(s) \in [0,1]^n$ and $P_{sa} \in [0,1]^n$ to represent the state transition probabilities for action $a$ in state $s$ and $P_a(i,j) \in [0,1]$ to represent the probability of going from state $i$ to state $j$ when taking action $a$.
- $\gamma \in [0,1]$ is the discount factor.
- $R : S \to \mathbb{R}$ is the reinforcement or reward function.

It is important to note that the state transition probability matrices are right stochastic. Mathematically this can be stated as $P_a(i,j) \geq 0 \ \forall \ i,j$ and $\sum_j P_a(i,j) = 1 \ \forall i$

In this paper the reward function is assumed to be a function of purely the state instead of the state and the action. This assumption is also made for the initial results in Ng & Russel (2000). A policy is defined as a map $\pi : S \to A$. Given a policy $\pi$, we can define two functions.

The *value function* at a state $s_1$ is defined as

$$V^\pi(s_1) = \mathbb{E}\big[R(s_1) + \gamma R(\theta(s_1)) + \gamma^2 R(\theta(\theta(s_1))) + \ldots \mid \pi \big]$$

where $\theta(s)$ represents the trajectory under policy $\pi$.
The *Q function* is defined as

$$Q^\pi(s,a) = R(s) + \gamma \mathbb{E}_{s' \sim P_a(s)}[V^\pi(s')]$$

The *Bellman Optimality equation* states that a policy $\pi^*(s)$ is an optimal policy for an MDP if and only if

$$\pi^*(s) \in \arg\max_{a \in A} Q^{\pi^*}(s,a), \quad s \in S$$

As shown in Ng & Russel (2000), for a finite-state MDP with reward $R$, and for $\pi^* \equiv a_1$, the Bellman optimality equation is equivalent to the following condition:

$$(P_{a_1}(i) - P_a(i))(I - \gamma P_{a_1})^{-1} R \geq 0 \quad \forall i = 1, \ldots, n; \ a \neq a_1$$

It can also be shown that $\pi^* \equiv a_1$ is the unique optimal policy if the above inequality is strict. We note that this condition is necessary and sufficient for the policy to be optimal for the reward. Thus,

this condition results in the fundamental constraint for standard MDP IRL problems. Further analysis of this condition is presented in the following section. In the subsequent sections, we use the notation

$$F_{ai} := (P_{a_1}(i) - P_a(i))(I - \gamma P_{a_1})^{-1}$$

The empirical maximum likelihood estimates of the transition probabilities from sampled trajectories are denoted by $\hat{P}_a(i)$ and in a similar fashion we use the notation

$$\hat{F}_{ai} := (\hat{P}_{a_1}(i) - \hat{P}_a(i))(I - \gamma \hat{P}_{a_1})^{-1}$$

By enforcing the Bellman optimality of the policy $\pi^*$, linear constraints on the reward function can be formed in the IRL problem. This leads to different formulations of the IRL problem including linear programming, Bayesian estimation, Maximum Weights for Apprenticeship Learning and Maximum Margin Planning. The IRL problem can be formed as an optimization problem by minimizing some loss function. For instance, one such formulation presented in Ng & Russel (2000) is as follows:

$$
\begin{aligned}
\underset{R}{\text{maximize}} \quad & \sum_{i=1}^{n} \min_{a \in \{a_2, \ldots, a_k\}} \left( \hat{F}_{ai}^T R \right) - \lambda \|R\|_1 \\
\text{subject to} \quad & \hat{F}_{ai}^T R \geq 0 \ \ \forall a \in A \setminus a_1, i = 1, \ldots, n \\
& \|R\|_\infty \leq R_{max}
\end{aligned}
\tag{2.1}
$$

The following norms are used throughout this paper. The infinity norm of a matrix $A = [a_{ij}]$ is defined as $\|A\|_\infty = \sup_{i,j} |a_{ij}|$. The $L^1$ norm of a vector is defined as $\|b\|_1 = \sum_i |b_i|$. The *induced matrix norm* is defined as $\|\|A\|\|_\infty = \sup_j \|a_j\|_1$ where $a_j$ is the $j$-th row of the matrix $A$. Note that for a right stochastic matrix $P$, we can see that $\|\|P\|\|_\infty = 1$ and $\|P\|_\infty \leq 1$.

## 3 Geometric analysis of the IRL problem

The objective of the Inverse Reinforcement Learning problem is to find a reward function that generates an optimal policy. As stated above, the necessary and sufficient conditions for a policy $\pi^*$ (without loss of generality $\pi^* \equiv a_1$) to be optimal are given by the Bellman Optimality principle and can be stated mathematically as

$$F_{ai}^T R \geq 0 \ \ \forall a \in A \setminus a_1, i = 1, \ldots, n$$

Clearly, $R = 0$ is always a solution. However this solution is degenerate in the sense that it also allows any and every other policy to be "optimal" and as a result is not of practical use. If the constraint of $R \neq 0$ is considered, then by noticing that the points $F_{ai} \in \mathbb{R}^n$, the set of reward functions generating the optimal policy $\pi_1$ is then the set of hyperplanes passing through the origin for which the entire collection of points $\{F_{ai}\}$ lie in one half space. The problem of Inverse Reinforcement Learning, then is equivalent to the problem of finding such a separating hyperplane passing through the origin for the points $\{F_{ai}\}$. Here we also assume none of the $F_{ai} = 0$ as this would mean that there is no distinction between the policies $\pi = a$ and $\pi_1 = a_1$.

This geometric perspective of the IRL problem allows the classification of all finite state, finite action IRL problems into 3 regimes, graphically visualized in Figure 1:

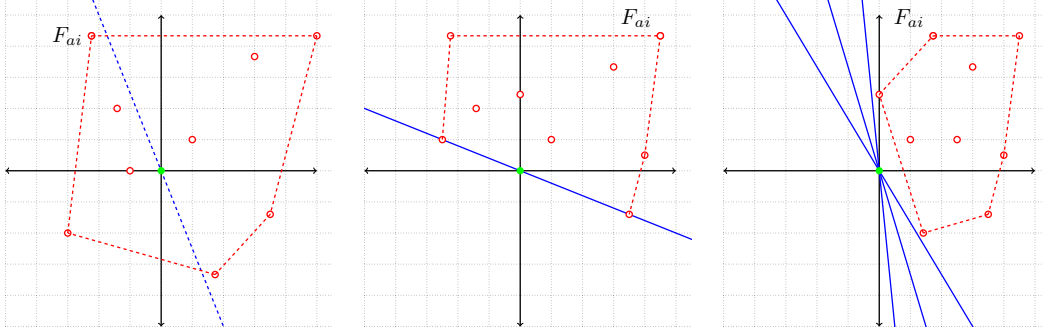

Figure 1: **Left:** An example graphical visualization of Regime 1 where the origin lies inside the convex hull of $\{F_{ai}\}$. Here no hyperplane passing through the origin exists for which all the points $\{F_{ai}\}$ lie in one half space. **Center:** An example graphical visualization of Regime 2 where the origin lies on the boundary of the convex hull of $\{F_{ai}\}$. Here only one hyperplane passing through the origin exists for which all the points $\{F_{ai}\}$ lie in one half space. **Right:** An example graphical visualization of Regime 3 where the origin lies outside the convex hull of $\{F_{ai}\}$. Here infinitely many hyperplanes passing through the origin exist for which all the points $\{F_{ai}\}$ lie in one half space.

**Regime 1**: In this regime, there is no hyperplane passing through the origin for which all the points $\{F_{ai}\}$ lie in one half space. This is equivalent to saying that the origin is in the interior of the convex hull of the points $\{F_{ai}\}$. In this case, independent of the algorithm, there is no nonzero reward function for which the policy $\pi_1$ is optimal.

**Regime 2**: In this regime, up to scaling by a constant, there can be one or more hyperplanes passing through the origin for which all the points $\{F_{ai}\}$ lie in one half space, however the hyperplanes always contain one of the points $\{F_{ai}\}$. This is equivalent to saying that the origin is on the boundary of the convex hull of the points $\{F_{ai}\}$ but is not one of the vertices since by assumption $F_{ai} \neq 0$. In this case, up to a constant scaling, there are one or more nonzero reward functions that generates the optimal policy $\pi_1$. In this case, it is also important to notice that the policy $\pi_1$ cannot be strictly optimal for any of the reward functions.

**Regime 3**: In this regime, up to scaling by a constant, there are infinitely many hyperplanes passing through the origin for which all the points $\{F_{ai}\}$ lie in one half space. This is equivalent to saying that the origin is outside the convex hull of the points $\{F_{ai}\}$. In this case, up to a constant scaling, there are infinitely many nonzero reward functions that generates the optimal policy $\pi_1$ and it is possible to find a reward function for which the policy $\pi_1$ is *strictly* optimal.

These geometric regimes and their implication on the finite state, finite action inverse reinforcement learning problem are summed up in the following theorem.

**Theorem 3.1.** *There exists a hyperplane passing through the origin such that all the points $\{F_{ai}\}$ lie on one side of the hyperplane (or on the hyperplane) if and only if there is a non-zero reward function $R \neq 0$ that generates the optimal policy $\pi = a_1$ for the inverse reinforcement learning problem $\{S, A, P_a, \gamma\}$. i.e., $\exists R$ such that $F_{ai}^T R \geq 0 \,\forall a, i$.*

**Remark 3.1.** *Notice that as an extension of Theorem 3.1, there is an $R$ for which the policy $\pi = a_1$ is strictly optimal iff there exists a hyperplane for which all the points $\{F_{ai}\}$ are strictly on one side.*

**Remark 3.2.** *Note that it is possible to find a separating hyperplane between the origin and the collection of points $\{F_{ai}\}$ if and only if the problem is in Regime 3. Therefore, the problem of inverse reinforcement learning can be viewed as a one class support vector machine (or as a two class support vector machine with the origin as the negative class) problem in this regime. This, along with the objective of determining sample complexity, leads in to the formulation of the problem discussed in the next section.*

## 4   Formulation of optimization problem

The objective function formulation of the inverse reinforcement problem described in Ng & Russel (2000) was formed by imposing the conditions that the value from the optimal policy was as far as

possible from the next best action at each state, as well as sparseness of the reward function. These were choices made by the authors to enable a unique solution to the proposed linear programming problem. We propose a different formulation in terms of a 1 class L1-regularized support vector machine that allows for a geometric interpretation as well as provides an efficient sample complexity. The Inverse Reinforcement Learning problem is now considered in Regime 3. Here it is known that there is a separating hyperplane between the origin and $\{F_{ai}\}$ so the strict inequality $F_{ai}^T R > 0$ which by scaling of $R$ is equivalent to $F_{ai}^T R \geq 1$. Formally this assumption is stated as follows

**Definition 4.1** ($\beta$-**Strict Separability**). *An inverse reinforcement learning problem* $\{S, A, P_a, \gamma\}$ *satisfies $\beta$-strict separability if and only if there exists a $\{\beta, R^*\}$ such that*

$$\|R^*\|_1 = 1 \quad and \quad F_{ai}^T R^* \geq \beta > 0 \quad \forall a \in A \setminus a_1, i = 1, \ldots, n$$

Notice that the IRL problem is in Regime 3 (i.e., $\exists w$ such that $w^T F_{ai} > 0$) if and only if the strict separability assumption is satisfied.

Strict nonzero assumptions are well-accepted in the statistical learning theory community, and have been used for instance in compressed sensing Wainwright (2009), Markov random fields Ravikumar et al. (2010), nonparametric regression Liu et al. (2008), diffusion networks Daneshmand et al. (2014).

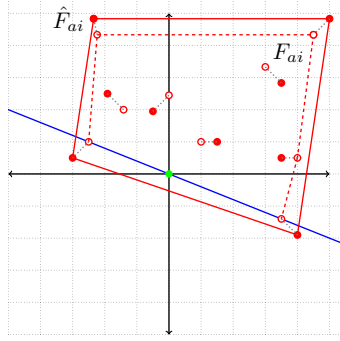

Figure 2: An example graphical visualization of Regime 2 the origin lies on the boundary of the convex hull of $\{F_{ai}\}$. Perturbation from statistical estimation of the transition probability matrices from empirical data (solid red), makes the problem easily tip into Regime 1 (shown) or Regime 3. An infinite number of samples would be required to solve IRL problems falling into Regime 2.

Problems in Regime 2 are avoided since based on the statistical estimation of the transition probability matrices from empirical data, the problem can easily tip into Regime 1 or Regime 3, as shown in Figure 2. To solve problems in Regime 2, an infinite number of samples would be required, where as problems in Regime 3 can be solved with a large enough number of samples.

Given the strict separability assumption, the optimization problem proposed is as follows

$$
\begin{aligned}
\underset{R}{\text{minimize}} \quad & \|R\|_1 \\
\text{subject to} \quad & \hat{F}_{ai}^T R \geq 1 \quad \forall a \in A \setminus a_1 \ \ i = 1, \ldots, n
\end{aligned}
\tag{4.1}
$$

This problem is in the form of a one class L1-regularized Support Vector Machine Zhu et al. (2004) except that we use hard margins instead of soft margins. The minimization of the $L^1$ norm plays a two fold role in this formulation. First, it promotes a sparse reward function, keeping in lines with the idea of simplicity. Second, it also plays a role in establishing the sample complexity bounds of the inverse reinforcement learning problem as well, as shown in the subsequent section. The constraints derive from strict Bellman optimality in the separable case (Regime 3) of inverse reinforcement learning and help avoid the degenerate solution of $R = 0$. We now use this optimization problem along with the objective of finding a reward function for which the policy $\pi = a_1$ is optimal to establish the correctness and sample complexity of the inverse reinforcement learning problem.

# 5 Correctness and sample complexity of Inverse Reinforcement Learning

Consider the inverse reinforcement learning problem in the strictly separable case (Regime 3). We have $\exists \{\beta, R^*\}$ such that

$$F_{ai}^T R^* \geq \beta > 0 \quad \forall a \in A \setminus a_1, i = 1, \ldots, n$$

Let $\|F_{ai} - \hat{F}_{ai}\|_\infty \leq \varepsilon$. Let $\hat{R}$ be the solution to the optimization problem 4.1 with $\hat{F}_{ai}$. We desire that

$$F_{ai}^T \hat{R} \geq 0 \quad \forall a \in A \setminus a_1, i = 1, \ldots, n$$

i.e., the reward we obtain from the problem using the estimated transition probability matrices also generates $\pi = a_1$ as the optimal for the problem with the true transition probabilities. This can be done by reducing $\varepsilon$, i.e., by using more samples. The result in the strictly separable case follows from the following theorem.

**Theorem 5.1.** *Let $\{S, A, P_a, \gamma\}$ be an inverse reinforcement learning problem that is $\beta$- strictly separable. Let $\hat{F}_{ai}$ be the values of $F_{ai}$ using estimates of the transition probability matrices such that $\|F_{ai} - \hat{F}_{ai}\|_\infty \leq \varepsilon$. Let $\hat{R}$ be the solution to the optimization problem 4.1 with $\hat{F}_{ai}$. Let $1 \geq c \geq 0$*

$$\varepsilon \leq \frac{1-c}{2-c}\beta$$

*Then we have $F_{ai}^T \hat{R} \geq c \quad \forall a \in A \setminus a_1, i = 1, \ldots, n$.*

**Remark 5.1.** *It is important to note that since $K, \beta > 0$ and $\varepsilon \geq 0$ and $c \leq 1 - \varepsilon\frac{K}{\beta}$, we have $c \leq 1$ with equality holding only when $\varepsilon = 0$, i.e infinitely many samples. This shows the equivalence of the problems with the true and the estimated transitions probabilities in the case of infinite samples.*

Our desired result then follows as a corollary of the above theorem.

**Corollary 5.1.** *Let $\{S, A, P_a, \gamma\}$ be an inverse reinforcement learning problem that is $\beta$- strictly separable. Let $\hat{F}_{ai}$ be the values of $F_{ai}$ using estimates of the transition probability matrices such that $\|F_{ai} - \hat{F}_{ai}\|_\infty \leq \varepsilon$. Let $\hat{R}$ be the solution to the optimization problem 4.1 with $\hat{F}_{ai}$.*

$$\varepsilon \leq \frac{1}{2}\beta$$

*Then we have $F_{ai}^T \hat{R} \geq 0 \quad \forall a \in A \setminus a_1, i = 1, \ldots, n$.*

*Proof.* Straightforwardly, by setting $c = 0$ in Theorem 5.1. $\qquad\square$

**Theorem 5.2.** *Let $\{S, A, P_a, \gamma\}$ be an inverse reinforcement learning problem that is $\beta$- strictly separable. Let the transition matrices $P_a$ have at most $d \in \{1, \ldots, n\}$ non-zero elements per row. Let every state be reachable from the starting state in one step with probability at least $\alpha$. Let $\hat{R}$ be the solution to the optimization problem 4.1 with $\hat{F}_{ai}$ with transition probability matrices $\hat{P}_a$ that are maximum likelihood estimates of $P_a$ formed from $m$ samples where*

$$m \geq \frac{64}{\alpha\beta^2}\left(\frac{(d-1)\gamma+1}{(1-\gamma)^2}\right)^2 \log\frac{4nk}{\delta}$$

*Then with probability at least $(1-\delta)$, we have $F_{ai}^T \hat{R} \geq 0 \quad \forall a \in A \setminus a_1, i = 1, \ldots, n$.*

The theorem above follows from concentration inequalities for the estimation of the transition probabilities, which are detailed in the following section. (All missing proofs are included in the Supplementary Material.)

# 6 Concentration inequalities

In this section we look at the propagation of the concentration of the empirical estimate of the transition probabilities around their true values.

**Lemma 6.1.** *Let A and B be two matrices, we have*

$$\|AB\|_\infty \le \|A\|_\infty \|B\|_\infty$$

Next we look at the propagation of the concentration of a right stochastic matrix $P$ to the concentration of its $k$-th power.

**Lemma 6.2.** *Let $P$ be a $n \times n$ right stochastic matrix with at most $d \in \{1, \ldots, n\}$ non-zero elements per row and let $\hat{P}$ be an estimate of $P$ such that*

$$\|\hat{P} - P\|_\infty \le \varepsilon$$

*then,*

$$\|\hat{P}^k - P^k\|_\infty \le ((k-1)d + 1)\varepsilon$$

Now we can consider the concentration of the expression $F_{ai} = (P_{a_1}(i) - P_a(i))(I - \gamma P_{a_1})^{-1}$.

Notice that since $P$ is a right stochastic matrix and $\gamma < 1$, we can expand $(I - \gamma P_{a_1})^{-1}$ as $(I - \gamma P_{a_1})^{-1} = \sum_{j=0}^{\infty} (\gamma P_{a_1})^j$ and therefore

$$(P_{a_1}(i) - P_a(i))(I - \gamma P_{a_1})^{-1} = (P_{a_1}(i) - P_a(i)) \sum_{j=0}^{\infty} (\gamma P_{a_1})^j$$

**Theorem 6.1.** *Let $P_a$ and $P_{a_1}$ be $n \times n$ right stochastic matrices with at most $d \in \{1, \ldots, n\}$ non-zero elements per row, corresponding to actions $a$ and $a_1$ and let $\gamma < 1$. Let $\hat{P}_a$ and $\hat{P}_{a_1}$ be estimates of $P_a$ and $P_{a_1}$ such that*

$$\|\hat{P}_a - P_a\|_\infty \le \varepsilon \quad \text{and} \quad \|\hat{P}_{a_1} - P_{a_1}\|_\infty \le \varepsilon$$

*Then, $\forall a, a_1 \in A$*

$$\left\| (\hat{P}_{a_1} - \hat{P}_a)(I - \gamma \hat{P}_{a_1})^{-1} - (P_{a_1} - P_a)(I - \gamma P_{a_1})^{-1} \right\|_\infty \le 2\varepsilon \frac{(d-1)\gamma + 1}{(1-\gamma)^2}$$

Note that this result is for each action. The concentration over all actions can be found by using the union bound over the set of actions.

An estimate of the value of $\varepsilon$ when the estimation is done using $m$ samples can be shown using the Dvoretzky-Kiefer-Wolfowitz inequality A. Dvoretzky & Wolfowitz (1956) to be on the order of $\varepsilon \in O\left( \sqrt{\frac{2 \log \frac{2n}{\delta}}{m}} \right)$.

This result is shown in the following Theorem 6.2.

**Theorem 6.2.** *Let $P_a$ be a $n \times n$ right stochastic matrix for an action $a \in A$ and let $\hat{P}_a$ be an maximum likelihood estimate of $P_a$ formed from $m$ samples. If $m \ge \frac{2}{\varepsilon^2} \log \frac{2n}{\delta}$, then we have*

$$\mathbb{P}\left[ \left\| \hat{P}_a - P_a \right\|_\infty \le \varepsilon \right] \ge 1 - \delta$$

The theorem above assumes that it is possible to start in any given state. However, this may not always be the case. In this case, as long as every state is reachable from an initial state with probability at least $\alpha$, the result presented in Theorem 5.2 can be modified to use Theorem 6.3 instead of Theorem 6.2.

**Theorem 6.3.** *Let $P_a$ be a $n \times n$ right stochastic matrix for an action $a \in A$ and let $\hat{P}_a$ be an maximum likelihood estimate of $P_a$ formed from $m$ samples. Let every state be reachable from the starting state in one step with probability at least $\alpha$. If $m \ge \frac{4}{\alpha\varepsilon^2} \log \frac{4nk}{\delta}$ then*

$$\mathbb{P}\left[ \left\| \hat{P}_a - P_a \right\|_\infty \le \varepsilon \right] \ge 1 - \delta, \ \delta \in (0,1) \forall a \in A$$

# 7 Discussion

The result of Theorem 5.2 shows that the number of samples required to solve a $\beta$-strict separable inverse reinforcement learning problem and obtain a reward that generates the desired optimal policy is on the order of $m \in O\left(\frac{d^2}{\beta^2}\log(nk)\right)$ for transition probability matrices with at most $d \in \{1, \ldots, n\}$ non-zero elements per row. Notice that the number of samples in inversely proportional to $\beta^2$. Thus by viewing the case of Regime 2 as $\lim \beta \to 0$ of the $\beta$-strict separable case (Regime 3), it is easy to see that an infinite number of samples are required to guarantee that the reward obtained will generate the optimal policy for the MDP with the true transition probability matrices.

In practical applications, however, it may be difficult to determine if an inverse reinforcement learning problem is $\beta$-strict separable (Regime 3) or not. In this case, the result of equation (A.1) can be used as a witness to determine that the obtained $\hat{R}$ satisfies Bellman's optimality condition with respect to the true transition probability matrices with high probability as shown in the following remark.

**Remark 7.1.** *Let $\{S, A, P_a, \gamma\}$ be an inverse reinforcement learning problem. Let the transition probability matrices $P_a$ each have at most $d \in \{1, \ldots, n\}$ non-zero elements per row. Let every state be reachable from the starting state in one step with probability at least $\alpha$. Let $\hat{R}$ be the solution to the optimization problem 4.1 with $\hat{F}_{ai}$ with transition probability matrices $\hat{P}_a$ that are maximum likelihood estimates of $P_a$ formed from $m$ samples and let*

$$\varepsilon = 2\sqrt{\frac{4}{\alpha m}\log\frac{4nk}{\delta}} \cdot \frac{(d-1)\gamma + 1}{(1-\gamma)^2}$$

*If $\|\hat{R}\|_1 \ll \frac{1}{\varepsilon}$, then with probability at least $(1 - \delta)$, we have $F_{ai}^T \hat{R} \geq 0 \quad \forall a \in A \setminus a_1, i = 1, \ldots, n.$*

# 8 Experimental results

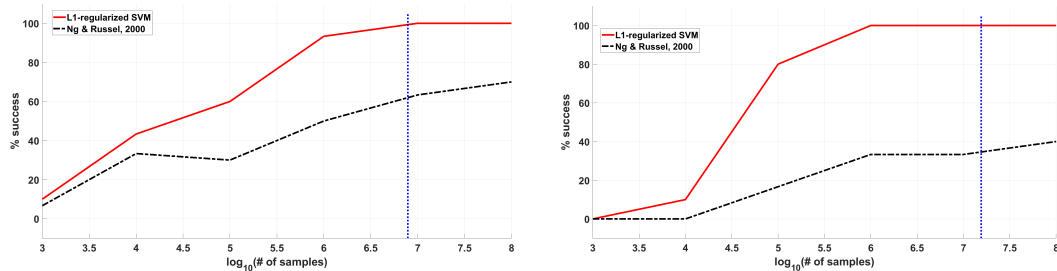

Figure 3: Empirical probability of success versus number of samples for an inverse reinforcement learning problem performed with $n = 5$ states and $k = 5$ actions (**Left**) and with $n = 7$ states and $k = 7$ actions (**Right**) using both our L1-regularized support vector machine formulation and the linear programming formulation proposed in Ng & Russel (2000). The vertical blue line represents the sample complexity for our method, as stated in Theorem 5.2

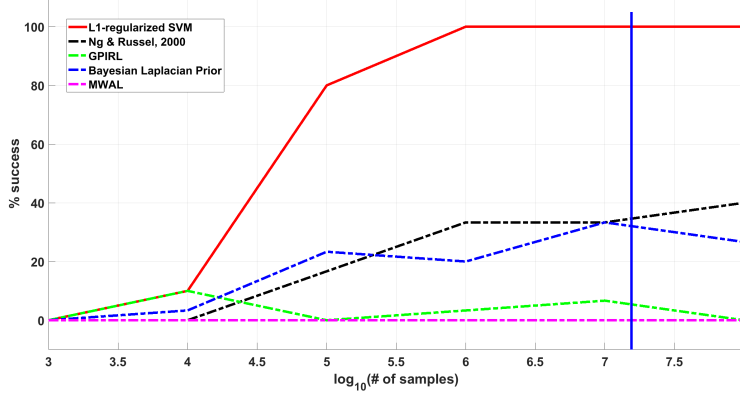

Figure 4: Empirical probability of success versus number of samples for an inverse reinforcement learning problem performed with $n = 7$ states and $k = 7$ actions using both our L1-regularized support vector machine formulation, the linear programming formulation proposed in Ng & Russel (2000), Multiplicative Weights for Apprenticeship Learning Syed et al. (2008) , Bayesian IRL with Laplacian prior Ramachandran & Amir (2007) and Gaussian Process IRL Levine et al. (2011). The vertical blue line represents the sample complexity for our method, as stated in Theorem 5.2

Experiments were performed using randomly generated transition probability matrices with $d = n$ non-zero elements per row, for $\beta$-strictly separable MDPs with $n = 5$ states, $k = 5$ actions, $\gamma = 0.1$ and with $n = 7$ states, $k = 7$ actions, $\gamma = 0.1$. Both experiments were done with $P_{a1}$ as the optimal policy. Thirty randomly generated MDPs were considered in each case and a varying number of samples were used to find estimates of the transition probability matrices in each trial. Reward functions $\hat{R}$ were found by solving Problem 4.1 for our L1-regularized SVM formulation, Problem 2.1 for the method of Ng & Russel (2000) along with code available at `https://graphics.stanford.edu/projects/gpirl/` to solve Multiplicative Weights for Apprenticeship Learning, Bayesian IRL with Laplacian prior and Gaussian Process IRL, using the same set of estimated transition probabilities, i.e., $\hat{F}_{ai}$. The resulting reward functions were then tested using the true transition probabilities for $F_{ai}^T \hat{R} \geq 0$. The percentage of trials for which $F_{ai}^T \hat{R} \geq 0$ held true is shown in Figure 3 and Figure 4 for different number of samples used. As prescribed by Theorem 5.2, for $\beta \approx 0.0032$, the sufficient number of samples for the success of our method is $O\left(\frac{n^2}{\beta^2} \log\left(nk\right)\right)$. As we can observe, the success rate increases with the number of samples as expected. The L1-regularized support vector machine, however, significantly outperforms the linear programming formulation proposed in Ng & Russel (2000), Multiplicative Weights for Apprenticeship Learning Syed et al. (2008), Bayesian IRL with Laplacian prior Ramachandran & Amir (2007) and Gaussian Process IRL Levine et al. (2011), reaching $100\%$ success shortly after the sufficient number of samples while the other methods fall far behind. The result is that the reward function given by the L1-regularized support vector machine formulation successfully generates the optimal policy $\pi = a_1$ in almost $100\%$ of the trials given $O\left(\frac{n^2}{\beta^2} \log\left(nk\right)\right)$ samples while the reward function estimated by the other methods fail to generate the desired optimal policy.

## 9    Concluding remarks

The L1-regularized support vector formulation along with the geometric interpretation provide a useful way of looking at the inverse reinforcement learning problem with strong, formal guarantees. Possible future work on this problem includes extension to the inverse reinforcement learning problem with continuous states by using sets of basis functions as presented in Ng & Russel (2000).

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
