[Supplementary Material]

This appendix contains the proofs for various Lemmas and Theorems presented in the paper.

## A   Proofs of Lemmas and Theorems

### A.1   Proof of Theorem 3.1

*Proof.* The proof follows from the fact that the points $\{F_{ai}\}$ lie on one side of the hyperplane passing through the origin given by $w^T x = 0$ if and only if

$$w^T F_{ai} \geq 0 \quad \forall a \in A \setminus a_1, i = 1, \ldots, n$$

or

$$w^T F_{ai} \leq 0 \quad \forall a \in A \setminus a_1, i = 1, \ldots, n$$

The proof in the 'if' direction follows by taking the hyperplane defined by $w = R$ and noticing that $w^T F_{ai} = R^T F_{ai} = F_{ai}^T R \geq 0$ so all the points $\{F_{ai}\}$ lie on one side of the hyperplane passing through the origin given by $R^T x = 0$

The proof in the 'only if' direction is as follows. Consider a separating hyperplane $w$. Without loss of generality,

$$w^T F_{ai} \geq 0$$

Now let $R = w$ then $F_{ai}^T R = R^T F_{ai} = w^T F_{ai} \geq 0$ so $R = w$ generates the optimal policy $\pi = a_1$. $\square$

### A.2   Proof of Theorem 5.1

*Proof.* Consider $F_{ai}^T \hat{R} \geq 0$, using Hölder's inequality we have

$$F_{ai}^T \hat{R} \geq -\|F_{ai} - \hat{F}_{ai}\|_\infty \|\hat{R}\|_1 + \hat{F}_{ai}^T \hat{R} \geq -\varepsilon \|\hat{R}\|_1 + 1 \tag{A.1}$$

Now let $\tilde{R} = \frac{K}{\beta} R^*$ where $K > 0$ and $R^*$ is the reward satisfying the $\beta$-strict separability for the problem. We have $\|\tilde{R}\|_1 = \frac{K}{\beta} \|R^*\|_1 = \frac{K}{\beta}$ as well as $F_{ai}^T \tilde{R} \geq K$. Now we have

$$\hat{F}_{ai}^T \tilde{R} \geq -\|F_{ai} - \hat{F}_{ai}\|_\infty \|\tilde{R}\|_1 + F_{ai}^T \tilde{R} \geq -\varepsilon \|\tilde{R}\|_1 + K = -\frac{K\varepsilon}{\beta} + K = K \left(1 - \frac{\varepsilon}{\beta}\right)$$

We now construct $\tilde{R}$ to satisfy the constraints of the optimization problem 4.1 with $\hat{F}_{ai}$ by choosing $K$ such that

$$\hat{F}_{ai}^T \tilde{R} \geq K \left(1 - \frac{\varepsilon}{\beta}\right) \geq 1 \implies K = \frac{1}{1 - \frac{\varepsilon}{\beta}}$$

Notice here since we have $K > 0$, then $\varepsilon < \beta$

Now since $\tilde{R}$ is a feasible solution to the optimization problem 4.1 with $\hat{F}_{ai}$ for which $\hat{R}$ is the optimal solution, we have from the objective function

$$\|\hat{R}\|_1 \leq \|\tilde{R}\|_1 = \frac{K}{\beta}$$

Substituting this upper bound for $\|\hat{R}\|_1$ in (A.1) we get,

$$F_{ai}^T \hat{R} \geq -\varepsilon \frac{K}{\beta} + 1 = 1 - \frac{\varepsilon}{\beta} \left(\frac{1}{1 - \frac{\varepsilon}{\beta}}\right) \geq 1 - \frac{1-c}{2-c} \left(\frac{1}{1 - \frac{1-c}{2-c}}\right) = 1 - \frac{1-c}{2-c}(2-c) = c$$

$\square$

### A.3 Proof of Theorem 5.2

*Proof.* The proof of this theorem is a consequence of Corollary 5.1 and Theorems 6.1 and 6.3. Note that from Theorem 6.3, we want the concentration to hold with probability $(1 - \delta)$ for all transition probability matrices corresponding to the set of actions. This can be viewed as the concentration inequality holding for a single $nk \times n$ matrix which gives us the result for $m$ samples

$$m \geq \frac{4}{\alpha \varepsilon^2} \log \frac{4nk}{\delta}$$

$$\implies \mathbb{P}\left[\left\|\hat{P}_a - P_a\right\|_\infty \leq \varepsilon_1\right] < 1 - \delta$$

The result then follows from substituting this value of $\varepsilon_1$ into the $\varepsilon$ in Theorem 6.1 and the consequent result into Corollary 5.1. $\square$

### A.4 Proof of Lemma 6.1

*Proof.* Let $C = AB$, we have $c_{ij} = \sum_k a_{ik} b_{kj}$

$$\|AB\|_\infty = \|C\|_\infty = \sup_{i,j} |c_{ij}|$$

From Holder's inequality we get

$$
\begin{aligned}
\|AB\|_\infty &= \sup_{i,j} \left\{ \left| \sum_k a_{ik} b_{kj} \right| \right\} \\
&\leq \sup_{i,j} \left\{ \left| \sum_k a_{ik} \right| \right\} \sup_{i,j} \{|b_{ik}|\} \\
&\leq \sup_{i,j} \left\{ \sum_k |a_{ik}| \right\} \sup_{i,j} \{|b_{ik}|\} \\
&= \|A\|_\infty \|B\|_\infty
\end{aligned}
$$

$\square$

### A.5 Proof of Lemma 6.2

*Proof.* First note that if $P$ is a right stochastic matrix then $P^k$ is a right stochastic matrix for all natural numbers $k$. Consider $n \times n$ right stochastic matrices $A, B, C, D$ with at most $d \in \{1, \ldots, n\}$ non-zero elements per row. Consider the expression $\|AC - BD\|_\infty$ From Lemma 1, we get,

$$
\begin{aligned}
\|AC - BD\|_\infty &= \|AC - AD + AD - BD\|_\infty \\
&\leq \|AC - AD\|_\infty + \|AD - BD\|_\infty \\
&\leq \|A\|_\infty \|C - D\|_\infty + \|A - B\|_\infty \|D\|_\infty
\end{aligned}
$$

Notice that $\|A - B\|_\infty \leq d\|A - B\|_\infty$ since $A$ and $B$ have at most $d \in \{1, \ldots, n\}$ non-zero elements per row, and $\|A\|_\infty = 1$ and $\|D\|_\infty \leq 1$, thus we have

$$\|AC - BD\|_\infty \leq \|C - D\|_\infty + d\|A - B\|_\infty$$

Now we will prove the lemma by induction $k = 1$. We have

$$\|\hat{P} - P\|_\infty \leq \varepsilon = ((1 - 1)d + 1)\varepsilon$$

Assume the statement for $k - 1$ is true. For $k > 1$ we have

$$\|\hat{P}^{(k-1)} - P^{(k-1)}\|_\infty \leq (((k-1)) - 1)d + 1)\varepsilon$$

Consider the previous result with $A = \hat{P}$, $B = P$, $C = \hat{P}^{(k-1)}$, $D = P^{(k-1)}$. Substituting, we get

$$\|\hat{P}\hat{P}^{(k-1)} - PP^{(k-1)}\|_\infty \leq (((k-1)) - 1)d + 1)\varepsilon + n\varepsilon$$

$$\implies \|\hat{P}^{(k)} - P^{(k)}\|_\infty \leq ((k-1)d + 1)\varepsilon$$

$\square$

## A.6 Proof of Theorem 6.1

*Proof.* Consider the expression from the theorem

$$\left\| (\hat{P}_{a_1} - \hat{P}_a)(I - \gamma \hat{P}_{a_1})^{-1} - (P_{a_1} - P_a)(I - \gamma P_{a_1})^{-1} \right\|_\infty$$

$$= \left\| (\hat{P}_{a_1} - \hat{P}_a) \sum_{j=0}^\infty \left( \gamma \hat{P}_{a_1} \right)^j - (P_{a_1} - P_a) \sum_{j=0}^\infty (\gamma P_{a_1})^j \right\|_\infty$$

$$= \left\| (\hat{P}_{a_1} - \hat{P}_a) \sum_{j=0}^\infty \left( \gamma \hat{P}_{a_1} \right)^j - \hat{P}_a \sum_{j=0}^\infty (\gamma P_{a_1})^j + \hat{P}_a \sum_{j=0}^\infty (\gamma P_{a_1})^j - (P_{a_1} - P_a) \sum_{j=0}^\infty (\gamma P_{a_1})^j \right\|_\infty$$

$$= \left\| \sum_{j=0}^\infty \gamma^j \left( \hat{P}_{a_1}^{j+1} - P_{a_1}^{j+1} \right) - (\hat{P}_a) \sum_{j=0}^\infty \gamma^j \left( \hat{P}_{a_1}^j - P_{a_1}^j \right) - (\hat{P}_a - P_a) \sum_{j=0}^\infty \gamma^j \left( P_{a_1}^j \right) \right\|_\infty$$

$$\leq \sum_{j=0}^\infty \gamma^j \left\| \left( \hat{P}_{a_1}^{j+1} - P_{a_1}^{j+1} \right) \right\|_\infty + \sum_{j=0}^\infty \gamma^j \left\| (\hat{P}_a) \left( \hat{P}_{a_1}^j - P_{a_1}^j \right) \right\|_\infty + \sum_{j=0}^\infty \gamma^j \left\| (\hat{P}_a - P_a) \left( P_{a_1}^j \right) \right\|_\infty$$

From Lemma 6.1 and Lemma 6.2; and the fact that for a right stochastic matrix $P$, $\|\|P\|\|_\infty = 1$ and $\|P\|_\infty \leq 1$; we have

$$\sum_{j=0}^\infty \gamma^j \left\| \left( \hat{P}_{a_1}^{j+1} - P_{a_1}^{j+1} \right) \right\|_\infty + \sum_{j=0}^\infty \gamma^j \left\| (\hat{P}_a) \left( \hat{P}_{a_1}^j - P_{a_1}^j \right) \right\|_\infty + \sum_{j=0}^\infty \gamma^j \left\| (\hat{P}_a - P_a) \left( P_{a_1}^j \right) \right\|_\infty$$

$$\leq \sum_{j=0}^\infty \gamma^j \left\| \left( \hat{P}_{a_1}^{j+1} - P_{a_1}^{j+1} \right) \right\|_\infty + \sum_{j=0}^\infty \gamma^j \left\| \left| \hat{P}_a \right| \right\|_\infty \left\| \left( \hat{P}_{a_1}^j - P_{a_1}^j \right) \right\|_\infty + \sum_{j=0}^\infty \gamma^j \left\| \left| \hat{P}_a - P_a \right| \right\|_\infty \left\| \left( P_{a_1}^j \right) \right\|_\infty$$

$$\leq \sum_{j=0}^\infty \gamma^j \left\| \left( \hat{P}_{a_1}^{j+1} - P_{a_1}^{j+1} \right) \right\|_\infty + \sum_{j=0}^\infty \gamma^j \left\| \left( \hat{P}_{a_1}^j - P_{a_1}^j \right) \right\|_\infty + \sum_{j=0}^\infty \gamma^j d \left\| \hat{P}_a - P_a \right\|_\infty$$

$$\leq \sum_{j=0}^\infty \gamma^j ((j)d + 1)\varepsilon + \sum_{j=0}^\infty \gamma^j ((j-1)d + 1)\varepsilon + \sum_{j=0}^\infty \gamma^j d\varepsilon$$

$$= \varepsilon \sum_{j=0}^\infty \gamma^j \left( (jd+1) + ((j-1)d+1) + d \right)$$

$$= 2d\varepsilon \sum_{j=0}^\infty j\gamma^j + 2\varepsilon \sum_{j=0}^\infty \gamma^j$$

$$= 2\varepsilon \left( \frac{d\gamma}{(1-\gamma)^2} + \frac{1}{1-\gamma} \right)$$

$$= 2\varepsilon \frac{(d-1)\gamma + 1}{(1-\gamma)^2}$$

$$\square$$

## A.7 Proof of Theorem 6.2

*Proof.* Here we invoke The Dvoretzky-Kiefer-Wolfowitz inequality A. Dvoretzky & Wolfowitz (1956). Consider $m$ samples of a random variable $Y_{ia}$ with domain $\{1, \ldots, n\}$, let $y_{ia}^{(1)}, \ldots, y_{ia}^{(m)} \in \{1, \ldots, n\}$ correspond to the observed resulting state under an action $a$ taken at a state $i$. Let $\hat{T}_{ia}(s) = \frac{1}{m} \sum_{j=1}^m \mathbb{1} \left[ y_{ia}^{(j)} \leq s \right]$ be an estimate of the CDF of $Y_{ia}$ and let $T_{ia}(s) = P[Y_{ia} \leq s]$ be the actual CDF. From the Dvoretzky-Kiefer-Wolfowitz inequality we have

$$\mathbb{P}\left(\sup_{s\in\{1,\ldots,n\}}\left|\hat{T}_{ia}(s)-T_{ia}(s)\right|>\varepsilon\right)\leq 2e^{-2m\varepsilon^2}$$

$$\implies\mathbb{P}\left(\sup_{s\in\{1,\ldots,n\}}\left|\hat{T}_{ia}(s)-T_{ia}(s)\right|\leq\varepsilon\right)>1-2e^{-2m\varepsilon^2}$$

Now consider the PDF of $Y_{ia}$ given by $\hat{\mathbf{p}}_{ia}(s)=\hat{T}_{ia}(s)-\hat{T}_{ia}(s-1)$. Notice that

$$|\hat{\mathbf{p}}_{ia}(s)-\mathbf{p}_{ia}(s)|\leq\left|\left(\hat{T}_{ia}(s)-\hat{T}_{ia}(s-1)\right)-(T_{ia}(s)-T_{ia}(s-1))\right|$$
$$\leq\left|\hat{T}_{ia}(s)-T_{ia}(s)\right|+\left|\hat{T}_{ia}(s-1)-T_{ia}(s-1)\right|$$

So if we have

$$\sup_{s\in\{1,\ldots,n\}}\left|\hat{T}_{ia}(s)-T_{ia}(s)\right|\leq\varepsilon$$

then

$$\sup_{s\in\{1,\ldots,n\}}|\hat{\mathbf{p}}_{ia}(s)-\mathbf{p}_{ia}(s)|\leq 2\varepsilon$$

$$\implies\mathbb{P}\left(\sup_{s\in\{1,\ldots,n\}}|\hat{\mathbf{p}}_{ia}(s)-\mathbf{p}_{ia}(s)|\leq\varepsilon\right)>1-2e^{-m\varepsilon^2/2}$$

Here we can interpret $\hat{\mathbf{p}}_{ia}(\cdot)$ and $\mathbf{p}_{ia}(\cdot)$ as the $i$-th rows of the matrices $\hat{P}_a$ and $P_a$ respectively. $\hat{\mathbf{p}}(Y_{ia})$, is the maximum likelihood estimator formed from $m$ samples. From application of the union bound over all rows of the matrix $P_a$, we have for $\varepsilon>0$, and $m$ samples,

$$\mathbb{P}\left((\forall i\in 1,\ldots,n)\,\|\hat{\mathbf{p}}(Y_{ia})-\mathbf{p}(Y_{ia})\|_\infty<\varepsilon\right)>1-2ne^{-m\varepsilon^2/2}$$
$$\implies\mathbb{P}\left[\left\|\hat{P}_a-P_a\right\|_\infty\leq\varepsilon\right]\geq 1-\delta,\ \delta\in(0,1)$$

if $m\geq\frac{2}{\varepsilon^2}\log\frac{2n}{\delta}$ $\qquad\qquad\qquad\qquad\qquad\qquad\qquad\qquad\qquad\qquad\qquad\square$

### A.8 Proof of Theorem 6.3

*Proof.* Without loss of generality, let every state $j=1,\ldots,n$ be reachable from state $j=1$ by action $a_1$ after a step with probability at least $\alpha$. Let $Y_{ja}$ be a random variable domain $\{1,\ldots,n\}$. Let $Z_j$ be a Bernoulli random variable such that $P(Z_j=1)\geq\alpha\forall j$. Let $(z_j^{(1)},y_j^{(1)}),\ldots,(z_j^{(m)},y_j^{(m)})$ be $m$ pairs of independent samples of $Z_j$ and $Y_{aj}$. Here $Z_j$ represents the state chain $1\xrightarrow{a_1} j\to\ldots$

Consider the event $A_1\equiv\{\frac{1}{m}\sum_{k=1}^{m}z_j^{(k)}\geq\alpha-\epsilon\forall j\}$. By the one-sided Hoeffding's inequality and taking the union bound over all states we have

$$\mathbb{P}(A_1)\geq 1-ne^{-2\epsilon^2 m}$$

We also have the conditional maximum likelihood probability estimator

$$\hat{\mathbf{p}}(Y_j=s|Z_j=1)=\frac{1}{\sum_{k=1}^{m}z_j^{(k)}}\sum_{l=1}^{m}\mathbb{1}[(y_j^{(l)}=s)\wedge z_j^{(l)}]$$

From Theorem 6.2 we have for event

$$A_2\equiv\{\|\hat{\mathbf{p}}(Y_{ja}|Z_j=1)-\mathbf{p}(Y_{ja}|Z_j=1)\|_\infty\leq\beta\}$$

$$\mathbb{P}(A_2|A_1)\geq 1-2ne^{-2\beta^2 m(\alpha-\epsilon)/2}$$

By the law of total probability

$$\mathbb{P}(A_2) \geq \mathbb{P}(A_2, A_1) = \mathbb{P}(A_2|A_1)\mathbb{P}(A_1)$$
$$= \left(1 - ne^{-2\epsilon^2 m}\right)\left(1 - 2ne^{-2\beta^2 m(\alpha - \epsilon)/2}\right)$$
$$\geq 1 - ne^{-2\epsilon^2 m} - 2ne^{-2\beta^2 m(\alpha - \epsilon)/2}$$

By solving $\frac{\delta}{2} = ne^{-2\epsilon^2 m}$ and $\frac{\delta}{2} = 2ne^{-2\beta^2 m(\alpha - \epsilon)/2}$ we can see that if $m \geq$ $\max\left\{\frac{1}{2\epsilon^2}\log\frac{2n}{\delta}, \frac{2}{(\alpha - \epsilon)\beta^2}\log\frac{4n}{\delta}\right\}$ then $\mathbb{P}(A_2) \geq 1 - \delta$ Letting $\epsilon = \frac{\alpha}{2}$ and taking the union bound over all actions $a \in A$ we have if $m \geq \frac{4}{\alpha\beta^2}\log\frac{4nk}{\delta}$ then

$$\mathbb{P}\left[\left\|\hat{P}_a - P_a\right\|_\infty \leq \beta\right] \geq 1 - \delta, \ \delta \in (0, 1)\forall a \in A$$

$\square$

## B  Experiment setup and additional results

Experiments were performed in MATLAB using randomly generated transition probability matrices for $\beta$-strictly separable MDPs with $n$ states, $k$ actions, $\gamma$.

We generate the rows of the transition probability matrices individually in one of two different ways. In the first method each row $P_a(i) = x \in [0,1]^n$ is generated as a uniformly sampled point from the region $\mathcal{C} := \{x \in \mathbb{R}^n \mid x \succeq 0, \|x\|_1 = 1\}$. In the second method, we wanted to simulate situations where taking an action at a state would lead to a few states (say $n_1$ states) with a large probability and the remaining $n - n_1$ states with a small probability. This is similar to a situation where the action chosen is followed with a large probability and where state jumps to a random state with a small probability. This is based on the idea of following the action with a large probability and randomly "exploring" with a small probability. This is similar to transition rules used by Ng & Russel (2000) in their experiment. Both of these methods were tested in generating the "true" transition probability matrices. The results shown in Figure 5 of this supplement were obtained using transition probability matrices generated by the first method. The results presented in Figure 3 in the main paper and in Figure 6 of this supplement were obtained for transition probability matrices generated by the second method. We also checked all generated transition probability matrices to ensure $\beta$-separability. The maximum likelihood estimates $\hat{P}_{ai}$ of these transition probability matrices were formed by sampling trajectories under the true transition probability matrices with the action chosen uniformly at random at each state. Several trajectories were formed, each with a random initial state to ensure that each state was reachable in the simulations.

Recall that $F_{ai} = (P_{a_1}(i) - P_a(i))(I - \gamma P_{a_1})^{-1}$ and $\hat{F}_{ai} = (\hat{P}_{a_1}(i) - \hat{P}_a(i))(I - \gamma \hat{P}_{a_1})^{-1}$. Reward functions $\hat{R}$ were found by solving our L1-regularized SVM formulation, and the method of Ng & Russel (2000), using the same set of estimated transition probabilities, i.e., $\hat{F}_{ai}$. The resulting reward functions were then tested using the true transition probabilities for $F_{ai}^T \hat{R} \geq 0$.

Additional results for 30 repetitions of $n = 5$ states, $k = 5$ actions, separability $\beta = 0.0032$, with transition probabilities generated using the first method are shown in Figure 5. Results for 20 repetitions of $n = 10$ states, $k = 10$ actions, separability $\beta = 0.0032$, with transition probabilities generated using the second method are shown in Figure 6.

Figure 5: Empirical probability of success versus number of samples for an inverse reinforcement learning problem performed with $n = 5$ states and $k = 5$ actions using both our L1-regularized support vector machine formulation and the linear programming formulation proposed in Ng & Russel (2000). The samples were generated using the first method as described in this supplement. The vertical blue line represents the sample complexity for our method

Figure 6: Empirical probability of success versus number of samples for an inverse reinforcement learning problem performed with $n = 10$ states and $k = 10$ actions using both our L1-regularized support vector machine formulation and the linear programming formulation proposed in Ng & Russel (2000). The samples were generated using the second method described in this supplement (i.e., the same method used in the main paper). The vertical blue line represents the sample complexity for our method