[Reviews · NeurIPS 2019]

Reviewer 1



Originality: The work stems from an original idea but a better acknowledgment of the last 20 years of work on IRL would have provided a more honest basis for situating the contribution. Quality: The paper is technically sound and the methodology for the results is well-founded however, it is unclear how the theorems apply to the presented work. It would be interesting to see the broader impact of the theorems by analyzing how other algorithms proposed for IRL are impacted by these theorems and how these relate to the experimental results proposed. A lot of the proofs could have been left to the Appendix, leaving more space for experiments and discussion. Clarity: The paper is clearly written and good readability. It adequately informs the reader about some of the limitations of the method. A more in-depth introduction of the F matrix with respect to value iteration would make the paper more self-contained. Significance: The theorems presented in the paper seem important to characterize the number of samples needed to achieve a certain bound in the Finite state space IRL case but it is unclear how this impacts current state-of-the-art algorithms.

Reviewer 2



The submission improves the theoretical supports for the IRL problem in Ng & Russel (2000), by providing a sample complexity and a sufficient condition for the correctness of the IRL problem. The results are mainly based on a geometric analysis on the IRL problem which contributes in two aspects: 1. provides the sufficient condition; 2. transforms the problem to a L_1-regularized SVM problem, whose theoretical characteristics have already been thoroughly studied. Therefore, to better evaluate the contributions of the submission, I have following two questions: 1. How strong is the definition 4.1 for the IRL problem. (While I admit definition 4.1 is weak enough for L_1 regularized methods, why this is true for IRL problems?) In other words, by only considering the Regime 3 case, how much do we lose? 2. It seems that the formulation here is now a kind of constrained formulation, especially compared with maximum entropy IRL methods [1], which are more often practically used. Therefore, can the analysis in the submission here contribute or generalized to more more practical methods? [1] Finn, Chelsea, Sergey Levine, and Pieter Abbeel. "Guided cost learning: Deep inverse optimal control via policy optimization." International Conference on Machine Learning. 2016. =================================================== The rebuttal of the authors is clear and convincing. I would like to increase my score to 6.

Reviewer 3



Summary: This paper first proposed a geometric analysis perspective for the IRL problem. Based on this observation, an L_1 regularized SVM formulation was proposed. Theoretical analysis on the sample complexity provided formal guarantees for the algorithm. The empirical performance was also tested. Comments: Originality: The theoretical analyses in this paper, including the geometric perspective, the SVM formulation and the sample complexity analysis, are technically straightforward to some degree. Significance: Although the technique in this paper is not that 'fancy', the work is important and valuable. The proposed algorithm is the first one with formal guarantees for the IRL problem. Quality: I think the theoretical analysis is fine. Empirical performance is also evaluated in the experiment part. Clarity: This paper is well-organized, and easy to follow. Other minor comments: Please go through the paper again to make the notations consistent, like in Eq (2.1), the capital N is not defined before. ========= Update after author response: The author didn't address my primary concern in detail, i.e., the technical challenges and novelty. But the theoretical contribution of their work is sufficiently significant for an acceptance. So I maintain my score.

Reviewer 4



This work introduces a geometric analysis of the problem of inverse reinforcement learning (IRL) and proposes a formal guarantee for the optimality of the reward function, obtained from the empirical data. The authors also provide the sample complexity for their proposed l1-regularized Support Vector Machine formulation. In general, this is an interesting work with a significant contribution to the theoretical aspect of the inverse reinforcement learning problem. However, there are a few concerns that need to be addressed: Major: 1. The paper does not define the problem as a stand-alone question in the field. The problem formulation heavily relies on the previous work by Ng & Russel (2000) and is written only as a follow up to this work. As such, the reader needs to go back and forth between the two papers to understand the terminology and contribution of this work. The authors need to clearly define the problem, provide the motivation, describe the previously proposed solutions, and clearly state their contribution. Specifically, the section “2. Preliminaries” which introduces Equation 2.1 is far from self-explanatory. 2. There needs to be more discussion about the previous work in inverse reinforcement learning. Many different techniques have been proposed during the past few years (Arora and Doshi, 2018), none of which are discussed and compared to here, which casts doubt on the novel contribution of the paper to the field. 3. While the theoretical aspect of the work is widely discussed, the practical aspect is not adequately addressed. In particular, three similar experiments with small state and action spaces were employed (n, k=5; n, k=7; n, k=10). The authors need to extend their analysis to provide a comparison of the two approaches in larger-scale environments with more realistic tasks (e.g., grid world, Atari game, navigation, etc.) Minor: a. The value function in section “2. Preliminaries” is not defined correctly. \pi: S -> A, and thus R(\pi(s)) is defined in the action space, whereas the reward function is assumed to be defined on the state space. Similarly, \pi(\pi(s)) is not meaningful because the policy is defined in the state space, not action space. The notations should change to resolve this confusion. b. All equations need to have numbers. c. In Equation 2.1, what is N? This should be n. d. Equation 4.1 should be a minimization rather than a maximization.

[Author Response · NeurIPS 2019]

Our paper presents an geometric interpretation for inverse reinforcement learning (IRL) with finite states and actions, as well as a corresponding L1-regularized Support Vector Machine formulation with formal guarantees in sample complexity and with regard to Bellman optimality. We thank the reviewers for their feedback and for bringing several issues and improvements to our attention. Our main contribution and focus in this paper is theoretical, which the reviewers seem to agree with, and the experiments serve primarily to validate our theory.

Following comments from the reviewers, we will move The proof of Theorem 5.1 to the appendix and add content discussing the background of IRL. In addition to the linear programming method [5] and the Bayesian IRL method [6] presented in the original submission, several other approaches to the IRL problem exist. These include Hybrid IRL [4], Maximum Margin Planning [7], Maximum Entropy IRL (MaxEnt) [9], and Gaussian Process IRL (GPIRL) [3]. A survey of other approaches to the inverse reinforcement learning problem can be found in [1]. While the problem in our paper is formulated in the form of a standard Markov Decision Process (MDP), several approaches instead consider the linearly-solvable MDP (LMDP) formulation presented in [2]. As mentioned in [2] (in particular Section 2.6), the problem formulation of the standard MDP and LMDP is different. Given the true transition probabilities, methods such as Multiplicative Weights for Apprenticeship Learning (MWAL) [8] and [5] are guaranteed to recover the true optimal policy where as methods that use LMDP to solve the same problem are not. To confirm this, we have used GPIRL [3] code available at `https://graphics.stanford.edu/projects/gpirl/` to solve the same synthetic experiments presented in Figure 3 of Section 8 of our paper and found the rewards from GPIRL were not Bellman optimal as per the definition in section 3 line 75 of our paper. Further comparisons with GPIRL, Bayesian IRL and MWAL are shown the Figure 1. We note that about $30\%$ of the rewards returned by MWAL were the trivial solution $R = 0$ which were not counted as successes. The result presented in our paper immediately impacts algorithms that use standard MDP models more it impacts than algorithms that use LMDP models such as MaxEnt and GPIRL, as the objective of the LMDP-based algorithms is different. In the case of standard MDP problems it readily provides a sample complexity and a formal guarantee with respect to Bellman optimality, which is not provided by any of the other methods.

The formulation provided in our paper is a nonparametric approach as compared to approaches that use features derived from states. It also makes no assumptions on the sparseness of the transitions. Our experiments reflect this as well, as the transition probabilities are drawn from a uniform distribution with no sparseness assumptions and would be more difficult to than sparse cases. In contrast, tasks like gridworld tend to have have sparse transitions probabilities or feature transformations that reduce dimensions. We provide a method and a guarantee on optimality that does not require sparseness and does not depend on feature selection, which is a problem with other methods.

Definition 4.1 leads to our paper considering only Regime 3 cases. From a practical perspective this is not a loss. All problems within Regime 1 have only one feasible solution, $R \equiv 0$ which does not provide any information with respect to Bellman optimality as no policy is preferred over the other with this reward. Problems in Regime 2 require an infinite number of samples to both ascertain that the problem is indeed in Regime 2 and to solve the problem as perturbations can lead to the estimated problem falling under Regime 1 or Regime 3 as mentioned in line 134 of our paper. As stated in line 130, this type of assumption has been made in other areas as well.

While our paper does state the parameters and objective of the finite state MDP IRL problem from lines 39-76 as well as an initial formulation, our paper does not aim to derive the results and formulation of [5] from scratch. As suggested, we will present this problem as a standalone problem and explain the origin of the F matrix from Bellman Optimality.

# References

[1] S. Arora and P. Doshi. A survey of inverse reinforcement learning: Challenges, methods and progress. *arXiv:1806.06877*, 2018.

[2] K. Dvijotham and E. Todorov. Inverse optimal control with linearly-solvable mdps. In *ICML 2010*, pages 335–342, 2010.

[3] Sergey Levine, Zoran Popovic, and Vladlen Koltun. Nonlinear inverse reinforcement learning with gaussian processes. In *NeurIPS 2011*, pages 19–27, 2011.

[4] Gergely Neu and Csaba Szepesvári. Apprenticeship learning using inverse reinforcement learning and gradient methods. In *UAI 2007*, pages 295–302. AUAI Press, 2007.

[5] A. Y. Ng and S.J. Russel. Algorithms for inverse reinforcement learning. In *ICML 2000*, pages 663 – 670, 2000.

[6] Deepak Ramachandran and Eyal Amir. Bayesian inverse reinforcement learning. *IJCAI 2007*, 51(61801):1–4, 2007.

[7] N. D. Ratliff, J. A. Bagnell, and M. A. Zinkevich. Maximum margin planning. In *ICML 2006*, pages 729–736. ACM, 2006.

[8] U. Syed, M. Bowling, and R. Schapire. Apprenticeship learning using linear programming. In *ICML 2008*, pages 1032–1039. ACM, 2008.

[9] Brian D Ziebart, Andrew Maas, J Andrew Bagnell, and Anind K Dey. Maximum entropy inverse reinforcement learning. *AAAI 2008*.

Figure 1:

[Meta-Review · NeurIPS 2019]

After discussions among the reviewers, they all agreed that the paper has enough theoretical contributions. They were also very happy with rebuttal and the answers provided by the authors. Please incorporate the suggested comments and references in the final version.